

# The COVID-19 pandemic "anthropause" decreased plastic ingestion in neotropic cormorants *Nannopterum brasilianus* in Lima, Peru

Laura Catalina Porras-Parra[1], Carlos B. Zavalaga[2] and Alvaro Rios[1]

[1] Unidad de Investigación de Ecosistemas Marinos-Grupo Aves Marinas. Carrera de Biología Marina, Universidad Cientifica del Sur, Lima, Peru

[2] Unidad de Investigación de Ecosistemas Marinos-Grupo Aves Marinas, Universidad Cientifica del Sur, Lima, Perú

## ABSTRACT

**Background**. The anthropause during the recent COVID-19 pandemic provided a unique opportunity to examine the impact of human activity on seabirds. Lockdowns in Peru prevented people from visiting coastal areas, thereby reducing garbage disposal on beaches and the movement of microplastics into the ocean. This cessation of activities likely led to a temporary decrease in plastic pollution in coastal regions. We aimed to investigate this phenomenon in inshore-feeding neotropic cormorants (*Nannopterum brasilianus*) along the Circuito de Playas Costa Verde (CPCV), situated on the coastal strip of Lima, Peru ($\sim$ 11 million people).

**Methods**. We collected and analyzed fresh pellets along the CPCV before (over 11 months) and during the pandemic lockdowns (over 8 months).

**Results**. Our findings revealed a significant reduction in the occurrence of plastic in pellets during the pandemic period (% Oc = 2.47, $n = 647$ pellets) compared to pre-pandemic conditions (% Oc = 7.13, $n = 800$ pellets). The most common plastic debris item found in the pellets was threadlike microplastic. Additionally, our study highlights the direct correlation between human presence on beaches and the quantity of microplastics (mainly threadlike) found in cormorant pellets. We suggest that the reintroduction of these materials into the sea, previously accumulated on the coast, is likely facilitated by the movement and activity of beachgoers toward the ocean.

# INTRODUCTION

During the COVID-19 pandemic, the term "anthropause" was coined to describe the global reduction in human activity. This unusual and temporary decline had a significant impact on the environment (*Rutz, 2022*). The decrease in human mobility provided a unique opportunity to study how fauna responds to such a reduction (*Bíl et al., 2021*; *Coll, Ortega-Cerdà & Marcarell-Roller, 2021*; *Madhok & Gulati, 2022*; *Markard & Rosenbloom, 2020*).

Corresponding author
Carlos B. Zavalaga,
czavalaga@cientifica.edu.pe

The lockdowns and curfews implemented during 2020–2021 provided an opportunity to study the effects of human activity on the environment by comparing conditions before, during, and after the pandemic in sites with varying levels of social mobility restrictions (*Rutz et al., 2020*). This unique opportunity allowed detailed analysis of the various interactions between humans and wildlife, which is crucial for the development of conservation policies (*Bates et al., 2021*). The diversity of studies on the effects of the anthropause on animals has increased due to the lockdowns worldwide, indicating that this period was crucial for many species affected by human activity (*Manenti et al., 2020*; *Perkins, Shilling & Collinson, 2021*; *Schrimpf et al., 2021*).

As a result of the anthropause, changes in animal behavior have been reported worldwide. The use of roads, airports, and recreational areas for resting and breeding by various animal taxa was commonly observed (*Manenti et al., 2020*; *Schrimpf et al., 2021*). The behavior responses varied depending on the species, type of human mobility, and time scale. For instance, birds which typically fed in residential gardens were seen less frequently because homeowners were more frequently using their backyards for recreation during lockdowns (*Madhok & Gulati, 2022*). Furthermore, changes in road use during lockdowns led to a significant reduction in wildlife mortality, with notable decreases in wildlife-vehicle collisions in countries like Estonia, Spain, Israel, and the Czech Republic, as well as a 50% reduction in hedgehog mortality on roads in areas of Poland (*Bíl et al., 2021*; *Łopucki et al., 2021*). The occupancy of beaches by wildlife was also affected during lockdowns. Crabs, endangered sea turtles, iguanas, and various species of seabirds had the opportunity to occupy areas that were frequently visited by tourists (*Soto et al., 2021*). However, the absence of control, surveillance, and management led to increased threats to native wildlife, due to the invasion of non-native species and illegal hunting in some locations (*Manenti et al., 2020*).

Among the groups of birds that have benefited from the anthropause are seabirds (*Hentati-Sundberg et al., 2021*; *Schrimpf et al., 2021*). For example, magnificent frigatebirds (*Fregata magnificens*), laughing gulls (*Leucophaeus atricilla*), and cormorants (Phalacrocoracidae) were commonly observed on urban beaches (*Soto et al., 2021*). However, other species were affected by the absence of human activity in tourist areas. A reduction in reproductive success in common murres (*Uria aalge*) was reported due to disturbance from white-tailed eagles (*Haliaeetus albicilla*) when tourists were not present in the murres' breeding areas (*Hentati-Sundberg et al., 2021*). Nonetheless, while studies of the effects of the anthropause have been conducted on some beaches in Latin America (*Soto et al., 2021*) marine pollution, particularly plastic contamination, remains a pressing concern worldwide. Lima, a coastal city with over 11 million inhabitants (*INEI, 2022*), is notably affected, presenting high levels of marine pollution (*Ayala, Cabrera & Quispe, 2007*; *Purca & Henostroza, 2017*; *Tapia et al., 2018*; *Gambini et al., 2019*). One of the most visited beaches in Lima is the Circuito de Playas Costa Verde (CPCV), which not only receives millions of visitors during the summer but is also with high levels of plastic contamination (*Blondet, Plaza-Salazar & Barona, 2023*), including microplastics (*De-la Torre et al., 2020*).

Despite such challenges, along a 16 km stretch of the coastline, CPCV offers areas for recreation, education, sports, and vehicle transit (*Majluf, 2014*; *ATUC, 2022*). A group

of 460 neotropic cormorants are common residents along the CPCV (*Lozano-Sanllehi & Zavalaga, 2021*). This species is susceptible to indirect plastic ingestion from its prey (*Azzarello & Vleet, 1987*), making it an indicator for measuring the anthropause's effect on marine plastic pollution. By analyzing regurgitated pellets with indigestible elements (*Barrett et al., 2007*), crucial information about the incidence of plastic in the diet of neotropic cormorants can be obtained (*Barrett et al., 2007*). However, there is a noticeable lack of such information in Peru.

On March 16, 2020, Peruvian authorities implemented the first mandatory COVID-19 lockdown, which endured for several months. Given the circumstances and the notable presence of microplastics on beaches (*Chen & Chen, 2020*; *Bayo, Rojo & Olmos, 2019*; *Retama et al., 2016*), coupled with the substantial foot traffic typically observed in these areas before the pandemic, it is likely that plastic materials were introduced into the sea through the movement of people between the beach and the water. Furthermore, tourists bathing in the sea contribute to microplastic pollution in coastal waters by shedding textile fibers from their clothing (*Akkajit et al., 2021*). We hypothesized that during the lockdown in Peru, plastic ingestion by neotropic cormorants would significantly decrease compared to before the lockdowns. Consequently, the objective of this study was to compare the occurrence of plastics in the pellets of neotropic cormorants on CPCV before and during the COVID-19 pandemic.

## MATERIALS & METHODS

### Study area

The CPCV is a 16-km narrow coastal strip located in Lima, Peru, that experiences high traffic of vehicles and people throughout the year. The main characteristics of the CPCV include beaches, parks, sports centers, restaurants, clubs, parking areas located adjacent to the beaches, and infrastructure for pedestrian and vehicular traffic. The highway has two to three lanes designated for vehicular traffic in both directions. The median strip contains public lighting poles and telephone cables that serve as perching sites for neotropic cormorants at four main locations along the highway (*Lozano-Sanllehi & Zavalaga, 2021*). A total of 1,447 neotropic cormorant pellets were collected from a specific section of this highway (collection started at −12.124764°S, −77.039605°W and ended at −12.122067°S, −77.044313°W, Fig. 1) in the district of Miraflores between October 27, 2019, and February 28, 2021.

### Definition of the pandemic phases

We examined the contents and the frequency of occurrence (%FO) of the pellets to identify plastic and other anthropogenic non-plastic debris before (pre-pandemic) and during (pandemic) the COVID-19 lockdowns in Lima, Peru. Likewise, to determine any changes in diet composition of cormorants between phases, we calculated the relative number of prey items (%NU) for each phase by sorting and counting the otoliths identified to species level in the pellets. Pre-pandemic phase refers to the time before March 16, 2020, when human activities in Peru and the CPCV were not restricted. Throughout the year, people frequented the beaches for leisure, with the highest number of visitors during the austral

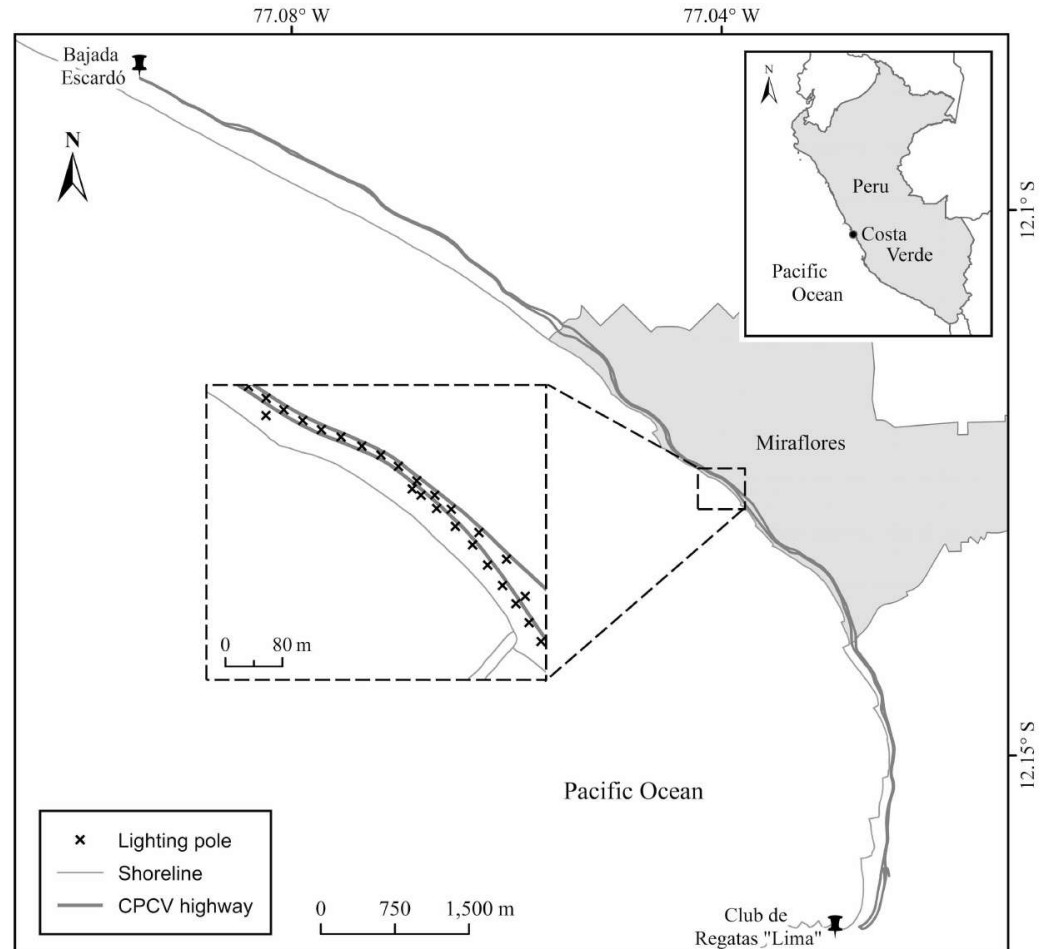

**Figure 1** **Map of the study area depicting part of the Circuito de Playas Costa Verde (CPCV) in the district of Miraflores.** The inset illustrates the specific section of the CPCV where the neotropic cormorant (*Nannopterum brasilianus*) pellets were collected.

summer (December–March). Despite the high influx of vehicles and people, neotropic cormorants were regularly seen perching on the light poles and telephone lines (*Majluf, 2014*; *Lozano-Sanllehi & Zavalaga, 2021*).

Before the pandemic, pellet collection was facilitated by a little wide area in the median strip located in the southern portion of the study area. During the pandemic, the restriction of vehicular circulation also allowed access to other areas along the narrow highway median for pellet collection. However, after the pandemic, the small area designated for pellet collection was closed, and vehicle access to the highway was restored, presenting a challenge to the continuation of pellet collection.

During the pandemic phase, access to the CPCV and nearby beaches was either completely prohibited (full lockdown) or restricted (partial lockdown) for people and vehicles (Table 1). Beaches were entirely off-limits to the public from March 16 to October

**Table 1** List of restriction measures on the Circuito de Playas Costa Verde (CPCV) highway and beaches during the Covid-19 pandemic phase in Peru.

| COVID-19 measure | Related decree | Dates | Restrictions on the CPCV |
| --- | --- | --- | --- |
| 1. Full lockdown | 044-2020- PCM, 080-2020-PCM and 1484-LD | 16 Mar.–1 Jul. 2020 | No access for people and vehicles to the beach and the highway. (1 Mayo: Reactivation of fisheries activities) |
| 2. Strict partial lockdown | 116-2020- PCM | 2 Jul.–22 Oct. 2020 | Access was limited to weekdays only, with no access permitted on weekends. Pedestrians were allowed access to the highway on weekends, but not to the beaches, except for non-contact watersports (*e.g.*, surfing, swimming). |
| 3. Partial lockdown | 170-2020-PCM and 184-2020- PCM | 23 Oct.–31 Dec. 2020 | Pedestrians were allowed partial access to the beach from Monday to Thursday. The highway and beaches were closed over the weekend. However, the beaches were open for non-contact water sports activities during weekdays. (25 Nov.-23 Dec.: The highway was also open on Sundays) (24, 25, and 31 Dec.: No public access to the highway and beaches) |
| 4. Strict partial lockdown | 202-2020-PCM, 002-2021-PCM, and 008-2021-PCM | 1 Jan.–24 Feb. 2021 | Access to the beaches only for non-contact watersports. The highway was only closed on Sundays from 5 am to 12.30 pm. (Second wave of COVID-19 in Perú.) |
| 5. End of lockdown | 208-2020- PCM | 25 Feb. 2021 | Entrance to the beaches and the highway was permitted. |

22, 2022. However, they were partially accessible from October 23, 2020, until February 28, 2021.

## Collection and analysis of pellets

During the pre-pandemic phase (October 2018–February 2020), a total of 11 visits were conducted. On each visit, a minimum of 11 and a maximum of 138 fresh pellets were collected as part of a project monitoring the neotropic cormorant diet. These pellets were stored in a freezer (−40 °C) for subsequent analysis.

During the pandemic phase (July 2020–February 2021) a total of 19 weekly visits were conducted to the study area, with between 11 and 77 fresh pellets being collected during each visit. All pellets were collected during the morning hours to obtain a greater quantity of fresh pellets, which were then individually packaged in 5 × 32 cm paper bags and frozen. Twenty-four hours before analysis, the pellets were defrosted in water and placed

on individual Petri dishes. All solid components (such as fish otoliths, vertebrae, shells, plastic, and others) were sorted and allowed to dry at room temperature. In this case, the otoliths found were considered to evaluate the changes and composition in the diet of the cormorants. After the sorting process, the otoliths were further washed and dried to remove any obstructing organic matter or debris. For the identification of the species pertaining to the otoliths we employed the use of the identification guides by *Oré-Villalba (2017)* and *García-Godos (2001)* as well as the UIEM (Unidad de Investigación de Ecosistemas Marinos) private otolith collection for reference.

For the plastic debris, if required, a stereoscope (10x magnification) was employed to classify specific plastic debris subcategories, as described by *Franeker et al. (2011)*:

- Sheetlike: Sheets derived from plastic bags.
- Threadlike: Filaments like ropes, threads, and plastic fibers.
- Fragment: Pieces of plastic objects such as bottles, boxes, toys, and tools.

The plastic debris (>0.1 mm) were measured using a digital caliper (Mitutoyo 150 mm, accuracy 0.1 mm). They were also classified according to their size into microplastics (5 mm), mesoplastics ($\geq$ 5 mm and $\leq$ 25 mm) and macroplastics (>25 mm). Additionally, non-plastic debris (>0.1 mm) such as glass, metal, paint chips, balloons, and fibers, were also measured in the same manner. All debris were grouped, counted, and separated according to their characteristics.

## Data analysis

During each sampling phase, the frequency of occurrence (%FO) of plastic was determined by dividing the count of pellets containing plastic debris by the total number of pellets collected, and then multiplying by 100. In some cases, pellets were collected on different days or weeks within a month. For data analyses, these collections were pooled in a single month. To test for differences in plastic intake by cormorants and prevalence of type of debris before and during the pandemic, a 2 ×2 Chi-square contingency table was used (*Silva-Costa & Bugoni, 2013*). Additionally, the average litter load in pellets was calculated by dividing the number of anthropogenic items found per month by the total number of pellets found per month. This approach facilitated the reporting of the average loads of plastic and other anthropogenic materials to determine if these loads also vary with lockdown levels. The Mann–Whitney $U$ test was used to compare the litter load per pellet between phases as data distribution did not follow the normality criteria (Shapiro–Wilks, $W = 0.564$, $P < 0.0001$). Changes in diet composition were tested with a paired $t$-test by comparing the %NU of the main prey.

All field procedures were ethically approved by the committee of ethics Universidad Cientifica del Sur (Constancia N ° 057-CIEI-AB-CIENTÍFICA-2021).

## RESULTS

### Pellet composition

Plastic debris and non-plastic debris were found in 96 out of 1,447 pellets analyzed (%FO = 6.63) during both the pre-pandemic and pandemic phases. The total number of debris

**Table 2  Composition of plastic and non-plastic debris found in 1,447 analyzed pellets of the neotropic cormorant, *N. brasilianus*, on the Circuito de Playas Costa Verde (CPCV), Lima, Perú, during both the pre-pandemic and pandemic phases.** The table provides a summary of anthropogenic categories, with their respective median sizes, size ranges, item counts, and percentage occurrence in pellets.

| Plastic categories | Median (mm) | Size (Range mm) | Items (n) | Pellets (n) | %FO (pellets) |
|---|---|---|---|---|---|
| Sheetlike | 6.26 | 28.91 | 24 | 21 | 1.45 |
| Threadlike | 13.63 | 200.61 | 96 | 35 | 2.42 |
| Fragments | 5.94 | 24.8 | 23 | 22 | 1.52 |
| **Rubbish categories** | **Median(mm)** | **Size(Range mm)** | **Items (n)** | **Pellets (n)** | **%FO (pellets)** |
| Glass | 7.70 | 6.68 | 3 | 3 | 0.21 |
| Wood | 12.09 | 7.49 | 3 | 3 | 0.21 |
| Metal | 10.67 | 14.56 | 4 | 4 | 0.28 |
| Fibers | 23.44 | 51.26 | 10 | 10 | 0.69 |
| Balloon | 9.71 | 0 | 1 | 1 | 0.07 |
| Paint chip | 2.15 | 1.07 | 2 | 2 | 0.14 |

items identified was 179, of which 150 were classified as plastic debris. Threadlike items (micro and mesoplastics) were the most common plastic debris category compared to sheetlike and fragment items (Table 2). The occurrence of threadlike items in pellets with debris items showed no significant difference before (%FO = 39%) and during (%FO = 56%) the pandemic period (Chi-square, $\chi^2 = 0.953$, $P = 0.329$, $df = 1$).

In addition to plastic, various other types of anthropogenic materials found in 23 out of 1,447 pellets were identified and classified as non-plastic debris. The most encountered category was fibers, followed by metal, glass, wood, paint chips and balloons (Table 2).

## Phases of COVID-19 and the presence of anthropogenic materials

During the pre-pandemic phase, 57 pellets (%FO = 7.13, $N = 800$) contained plastic, whereas the content of 13 pellets (%FO = 1.62, $N = 800$) was identified as non-plastic debris. In the pandemic phase, 16 pellets (%FO = 2.47, $N = 647$) contained plastic debris, and no pellets contained non-plastic debris. This difference was significant (Chi-squared = 16.47, $df = 1$, $p$-value <0.0001, Fig. 2). There was a clear fluctuation in the percentage occurrence of plastic in the cormorant pellets both prior and during the COVID-19 pandemic (Fig. 2), but when data is grouped within seasons of the year (spring, summer and winter, no data for autumn), no significant differences in the percentage of occurrence of plastic in cormorant pellets were found during pre-pandemic (Chi-square = 0.052, $df = 1$, $p$-value = 0.974) nor during pandemic phases (Chi-square = 1.38, $df = 1$, $p$-value = 0.5).

During the pre-pandemic phase, there was a significantly higher overall litter load per pellet (Mann–Whitney $U$-test, $U = 71$, $p$-value = 0.028, Fig. 3). In February 2021, there was a peak of litter load attributed to the presence of one exceptional single pellet containing 57 litter items; apart from this anomaly, the litter load during this period was consistently low.
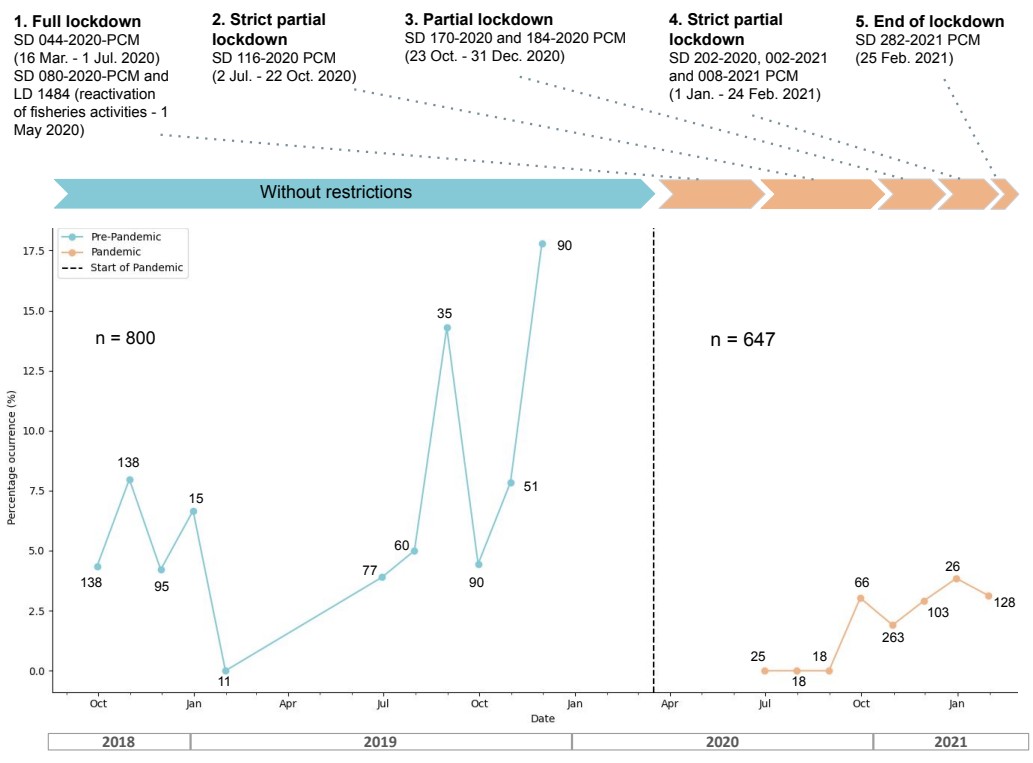

**Figure 2** **Overall monthly percentage of occurrence of plastic in the pellets of *N. brasilianus* during the pre-pandemic (blue) and pandemic (orange) phases of COVID-19 on the Circuito de Playas Costa Verde (CPCV), Lima, Perú.** The numbers along the lines indicate the number of pellets collected each month. N = total number of pellets collected prior and during the pandemic. The top legend describes the restrictions imposed by government authorities during the pandemic.

The diet of neotropic cormorants primarily consisted of five fish species, which accounted for 99% and 97% of the total prey species by number before and during the pandemic, respectively. There was no significant difference in the type of prey and the percentage numerical abundance (%NU) between the two phases (Paired $t$-test, $t = 0.0762$, $df = 4$, $P = 0.24$). The main fish prey included big nose anchovy (*Anchoa nasus*), which comprised 56% before the pandemic and 35% during the pandemic, minor stardrum (*Stellifer minor*) at 20% *vs.* 23%, Peruvian anchovy (*Engraulis ringens*) at 10% *vs.* 17%, lorna drum (*Sciaena deliciosa*) at 9% *vs* 16%, and Peruvian silverside (*Odontesthes regia*) at 4% *vs.* 6%.

In terms of the classification of plastics by size during the analyzed periods, it was observed that during the pre-pandemic phase, 17.19% were macroplastics, 48.44% were mesoplastics, and 34.38% were microplastics. However, during the pandemic, these percentages shifted to 17.33% for macroplastics, 73.33% for mesoplastics, and 9.33% for microplastics. As mentioned previously, regarding the classification by type of plastic, it is important to clarify that no significant differences were found between the evaluated periods. However, the most common type of plastic observed throughout the study was threadlike (Fig. S4).

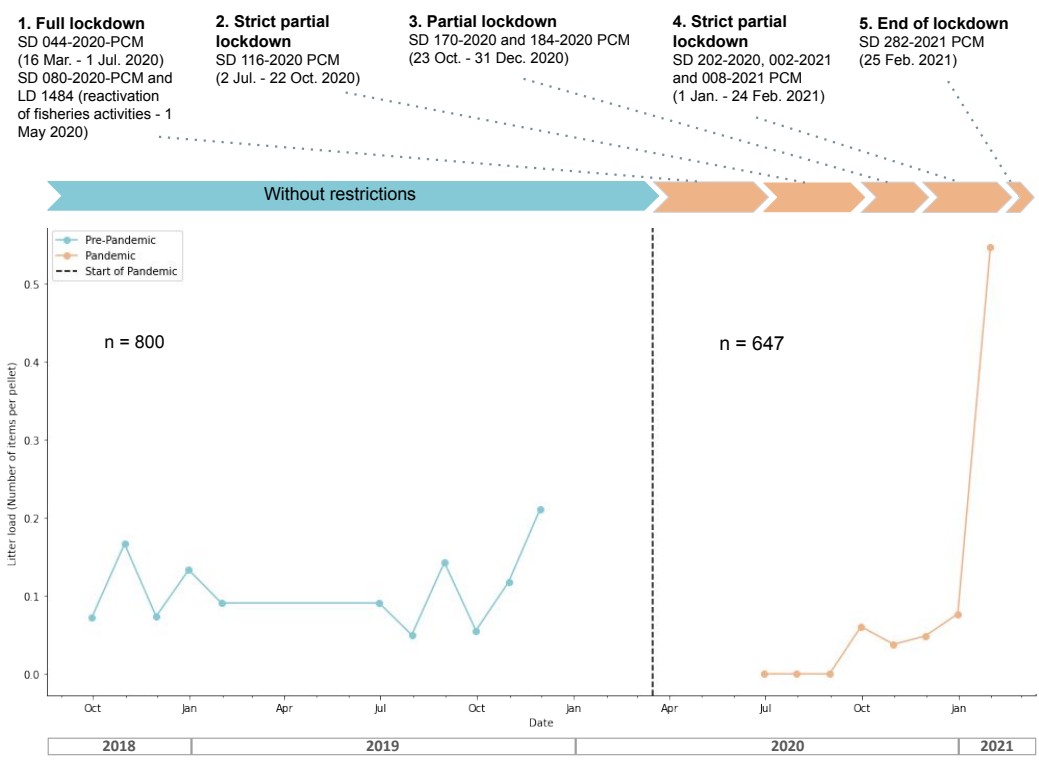

**Figure 3** Overall monthly litter load (average number of elements per pellets that measure the number of plastic and other anthropogenic items) of plastic items and non-plastic items in the pellets of *N. brasilianus* during the pre-pandemic (blue) and pandemic (orange) phases of COVID-19 on the Circuito de Playas Costa Verde (CPCV), Lima, Perú. The top legend describes the restrictions imposed by government authorities during the pandemic.

## DISCUSSION

This study revealed that the occurrence of plastic in neotropic cormorant pellets from CPCV beaches, Lima's most recreational coastal areas, was three times lower and had a higher litter load per pellet during a period of COVID-19 lockdowns in Peru compared to pre-pandemic conditions. The reduction in plastic ingestion is likely attributed to restrictions on beach access for beachgoers. Neotropic cormorants did not shift the type or relative abundance of prey consumed between the two phases. Therefore, any differences observed in the occurrence of plastic cannot be attributed to a change in diet if secondary plastic ingestion may have occurred. A lower volume of microplastic in the marine ecosystem seems to be an immediate response to a reduction of littering on beaches during the mandatory lockdowns (*Zambrano-Monserrate, Ruano & Sanchez-Alcalde, 2020*; *Okuku et al., 2021*; *Orzama-González, Castro-Rodas & Statham, 2021*; *Soto et al., 2021*), but microplastic found in cormorant pellets probably needed years to decades to breakdown from their original source (*Gregory, 2009*; *Andrady, 2011*), remaining in the marine environment even after the onset of lockdowns. However, the response by cormorants in reducing plastic ingestion was almost immediate, which suggests that other processes may be involved. Large amounts of

plastic particles had been already detected on the CPCV (*Blondet, Plaza-Salazar & Barona, 2023*) and microplastic debris in other beaches in Lima (*Purca & Henostroza, 2017*; *De-la Torre et al., 2020*) prior to the COVID-19 pandemic.

Two possible explanations for the quick turnover observed could be attributed to the introduction of microplastics into the sea due to the high presence of people engaging in recreational activities. These activities, such as swimming and surfing, have been shown to generate microplastic fibers through the shedding of clothing fibers (*Retama et al., 2016*; *Akkajit et al., 2021*; *Luo et al., 2022*), categorized as "threadlike" in this study, which were the most common type of microplastic and mesoplastic identified throughout the course of the study. Alternatively, it is suggested that microplastics were no longer displaced into the sea by beachgoers during their recreational activities when access was restricted. It is worth noting that sandy beaches may experience a greater reintroduction of microplastics, as sand can easily contain this type of residue (*Vermeiren et al., 2021*), which can then be transported to the sea, particularly critical in beaches adjacent to the collection site. These factors may also explain the progressive increase in plastic presence in pellets when restrictions on beach access were relaxed months after the onset of the pandemic. Additionally, oceanographic processes such as local currents and upwelling further influence the dynamics of plastic distribution in coastal areas. Region-specific oceanographic processes, as highlighted by *Manay et al. (2021)*, contribute to the accelerated presence of microplastics in coastal environments (*Lebreton, Greer & Borrero, 2012*; *Eriksen et al., 2014*).

Neotropic cormorants are considerably affected by entanglement in ghost nylon nets and the use of plastic for their nests (*Ayala et al., 2023*). Plastic has already permeated the marine trophic web in Peru, with reports in fish (*De-la Torre et al., 2019*; *Fernández & Anastasopoulou, 2019*), seabirds (*Thiel et al., 2018*; *Díaz-Santibañez, Clark & Zavalaga, 2023*), and marine mammals (*Perez-Venegas et al., 2020*; *Santillán, Saldaña Serrano & De-la Torre, 2020*). Given the common occurrence of threadlike plastic in marine sediment (*Cisneros, Montero & Guevara, 2021*), it is crucial to recognize its impact on the marine ecosystem. Being benthic divers (*Quintana et al., 2004*), neotropic cormorants consume benthic and mesopelagic fish (*Galarza, 1968*; *Casaux et al., 2009*; *Petracci et al., 2009*; *Muñoz Gil et al., 2012*). As a result, they are likely to ingest this type of plastic through trophic transfer from their prey. This finding is consistent with *Franco et al. (2019)* and *Álvarez, Barros & Velando (2018)*, who noted that threadlike plastic was the most frequently encountered category of plastic debris among seven seabird species. Nevertheless, the possibility of direct ingestion of plastic particles during underwater prey pursuit cannot be ruled out, considering their potential persistence within the water column—whether suspended, floating at the sea surface, or buried in the bottom sediment. This is particularly relevant as neotropic cormorants exhibit a foraging behavior that involves the exploitation of mid-water and bottom-dwelling fish (*Casaux et al., 2009*; *Petracci et al., 2009*). The observed high density of plastic fragments and microplastics on the beaches of the CPCV (*Blondet, Plaza-Salazar & Barona, 2023*) and neighboring coastal areas (*Purca & Henostroza, 2017*; *Gambini et al., 2019*) implies that the transport of such materials to the sea is likely facilitated by tidal movements, ocean swells, and human activities along the beaches.

Overall, plastic items were found in 7.13% of neotropic cormorant pellets during the pre-pandemic phase. These results are very similar to those reported by the Guanay cormorant in the coast of Peru (7% frequency of occurrence) (*Díaz-Santibañez, Clark & Zavalaga, 2023*) and within the range reported in other cormorant species (*Robards, Piatt & Wohl, 1995*; *Acampora, Newton & O'Connor, 2017*; *O'Hanlon et al., 2017*; *Brookson et al., 2019*; *Franco et al., 2019*; *Baak et al., 2020*). While our study provides insights into the immediate effects of lockdown measures on plastic ingestion by neotropic cormorants, it is crucial to acknowledge the complexity of plastic pollution dynamics, influenced by various factors. These factors include but are not limited to, human activities unrelated to the pandemic, such as the impact of wastewater treatment plants (WWTPs). Currently, only one WWTP, La Chira, is responsible for wastewater treatment in southern Lima, yet there is insufficient evidence to confirm its efficacy in waste eradication (ATUC, 2023). Moreover, WWTPs do not fully eliminate microplastics, with removal rates sometimes falling below 30% (*Lyare & Bond, 2020*; *Xu, Bai & Ye, 2021*). Thus, the significant influence of WWTPs on the occurrence of microplastics and mesoplastics in the sea near CPCV cannot be overlooked. Future research endeavors should delve deeper into this aspect to comprehensively understand how human activities and environmental factors collectively contribute to plastic pollution and its uptake by marine organisms.

## CONCLUSIONS

In this study, a decrease in the frequency of plastic occurrence in neotropical cormorant pellets was observed, from 7.13% in the pre-pandemic phase to 2.47% during the COVID-19 pandemic. This highlights the potential impact of beach access restrictions on reducing marine plastic pollution. However, the presence of plastics was not completely eradicated even with reduced human activity, suggesting the need for further research to fully understand the various factors that contribute to this type of pollution, such as WWTPs. Additionally, a post-pandemic examination of neotropical cormorant pellets could have provided a more complete understanding of anthropogenic plastic pollution. Unfortunately, post-pandemic access to the pellet collection area was restricted due to inaccessibility or safety concerns, which limited their collection. Despite this limitation, our findings indicate that neotropical cormorant pellets can be an effective non-invasive tool for monitoring plastic pollution in marine environments.

## ACKNOWLEDGEMENTS

This study was conducted during my final year of undergraduate studies in Marine Biology, as I sought immersion in the scientific realm through the exploration of *ad-hoc* hypotheses, meticulous data analysis, and proficient scientific writing. The authors express sincere gratitude to all members of the Unidad de Investigación de Ecosistemas Marinos-Grupo Aves Marinas, with special appreciation for Isabella Díaz-Santibañez for their invaluable assistance and collaboration in executing this research. Furthermore, our thanks extend to Sebastián Lozano-Sanllehi for his noteworthy contributions to editing the study area map. A special acknowledgment goes to Michał Czapliński for his assistance in pellet collection.

### Funding

This work was supported by Universidad Científica del Sur to the project code 083-2021-PREB5. The Universidad Científica del Sur also provided funding for the article processing fee. The funders had no role in study design, data collection and analysis, decision to publish, or preparation of the manuscript.

### Grant Disclosures

The following grant information was disclosed by the authors:
Universidad Científica del Sur:  083-2021-PREB5.

### Competing Interests

The authors declare there are no competing interests.

### Author Contributions

- Laura Catalina Porras-Parra performed the experiments, analyzed the data, prepared figures and/or tables, authored or reviewed drafts of the article, and approved the final draft.
- Carlos B. Zavalaga conceived and designed the experiments, performed the experiments, analyzed the data, authored or reviewed drafts of the article, and approved the final draft.
- Alvaro Rios performed the experiments, analyzed the data, authored or reviewed drafts of the article, and approved the final draft.

### Field Study Permissions

The following information was supplied relating to field study approvals (i.e., approving body and any reference numbers):

Data collection were approved by the Ethic Committee at Universidad Científica del Sur with a permit number No. 057-CIEI-AB-CIENTÍFICA-2021.

### Data Availability

The data is available in the Supplementary Files and figshare: Zavalaga, Carlos (2023). Datos bolos N. brasilianus.xlsx. figshare. Dataset. https://doi.org/10.6084/m9.figshare.23226416.v1.

### Supplemental Information

Supplemental information for this article can be found online at http://dx.doi.org/10.7717/peerj.17407#supplemental-information.

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
