# Peer review of "The COVID-19 pandemic “anthropause” decreased plastic ingestion in neotropic cormorants Nannopterum brasilianus in Lima, Peru"

_PeerJ, doi:10.7717/peerj.17407_

## Round 0.1 · original submission · Major Revisions

· Academic Editor

Major Revisions

Your article has now been seen by two reviewers and the reviewers’ comments are appended below. I share the reviewers' view that the data described in this article has the potential to provide valuable insights into how human littering might affect animal plastic consumption. However, I also agree with the concerns raised by both reviewers that the article currently does not sufficiently consider the processes through which the plastic items end up being ingested by the cormorants, which means that I do not think that this article is at this stage ready for publication. Both reviewers provide detailed comments that might help you to address what exactly might have influenced the patterns in your data. Both reviewers also provide additional requests for clarification to help readers better understand your study. I am looking forward to a revised manuscript.

Reviewer 1 ·

Excellent Review

This review has been rated excellent by staff (in the top 15% of reviews)
EDITOR COMMENT
This reviewer provided a balanced assessment, focusing on both the promise of the study and its limitations. The feedback was constructive, suggesting ways to the authors to address the limitations given the data at hand. The reviewer assessed all aspects of the study, including the supplementary data. As editor, I am grateful for such reviews because their thoroughness will not only hopefully help the authors to improve their study, but it also helps me as non-expert to make an assessment of the manuscript.

Basic reporting

The figures are good and appropraite. Figures 2 and 3 are really useful but they could be included as supplementary material if needed to save room as they are not critial to the manuscript. Table 2 and 3 would benefit from being combined into a single table.

Please check the supplementary table thoroughly. There are a few inconsistencies, i.e:
Row 663 & 789, 889, 923, 990, 1240 etc. Thread is including in the Classification general column not the plastic column
Row 996, 1313. Paint flakes are anthropogenic debris so the Count MA column should include a 1 – or if plastic, include that in the Classification general column
Row 1071. It’s not clear what ‘47.06/44.30/56.07 =31.92/24.15 ne’ means. Please clarify including in your Code Book tab.
Row 1423. Not clear what ‘Globo’ is
Row 81. Here wood is classified as an anthropogenic debris items but not on lines 159, 175, 178 etc
Row 107. 1 is wrong row (above) for glass in the Count MA column

Experimental design

The brief aims of the study should be formulated into a more specific hypothesis. For example, from the introduction it would be natural to assume more ingested plastics before compared to within the pandemic. The results can then be framed to specifically reflect your hyposthesis.

Validity of the findings

This is a really neat study taking good opportunity of the ‘experiment’ of the COVID pandemic. The conclusions are also important in how this highlights that actions to change people’s behaviour, in terms of plastic pollution, as well as improved waste management infrastructure can made a noticeable difference.

I do however, have a couple of concerns that need addressing.

1. Firstly, given that anthropogenic debris will be ingested by cormorants indirectly via their fish prey there is likely to be a time lag before any discarded plastics enter the ocean and is consumed by the fish. Furthermore, most fish of the size that cormorants eat will be ingesting small / mircoplastics i.e. degraded from larger discarded plastic items. Therefore, it seems very unlikely that the plastics found in the pellets reflect the situation at the time of collection. I realise that this time lag is difficult to quantify but it should be acknowledged and discussed in terms of how this will affect your results.

2. Much threadlike plastics in the ocean likely comes from fishing activities i.e. from degraded ropes and nets, rather than from consumer plastics associated with tourists. Do you have any information on the restrictions to fishers during the pandemic phases? This would strengthen your results/concusions. It should be discussed that fishing activities may / is likely be the source of these threadlike plastics – although again with a time lag to when they might be available and therefore appear in pellets via fish.

3. Figure 4 demonstrates that even within both pandemic phases there is variation over time in how much plastics are found in pellets. This needs to be discussed, and ideally accounted for in your analysis. Do you know why more plastics were found in Jan/ Feb than June/July for example. It would also be useful to know how many pellets were collected during each month during both pandemic phases to ensure that the difference seen is not influenced for example, by more pellets collected July in the pandemic phase – or associated with other unmeasured factors.

Additional comments

General comments.
Line 32. This sentence comes as a surprise as these sources of plastic pollution are not mentioned elsewhere in the manuscript.
Introduction: The first paragraph is a little repetitive – could be more concise to get the key point across.
Line 63. Place the relevant reference at the end of this sentence.
Line 64. Resident and rare species - not sure the relevance here as they may also be opportunistic?
Line 68-71. These sentences are very detailed, the message would be stronger if these two sentences were combined to make the broad comment that changes in road use affected wildlife mortality.
Line 76. Add 'in some locations'
Line 87. This is quite a sudden link to Peru - you could instead first move the focus to pollution, especially plastic pollution. And then link to Peru in being an area which is heavily impacted in specific areas.
Line 101. As well as including a stronger hypothesis here it should also be more explicitly stated why looking at cormorant pellets is useful. i.e. that the plastics / debris in the pellets reflects their presence in the environment. So the pellets can be sampled instead of the environment to detect changes in policies / people’s behaviour ect. However, they is also a large assumption made here that what the fish and therefore cormorant ingests and it subsequently found in the pellets, is representative in the environment. This is not always the case, especially, where species show a selection for specific plastics. Therefore this should be acknowledged in the discussion.
Line 114. Not clear what you mean by median here? The center of the highway?
Line 115. Recommend rephrasing this sentence for clarification by deleting ‘and pellet building-up sites’ and instead adding ‘where pellets build up’ at the end of the sentence.
Line 127. Add ‘anthropogenic’ before debris.
Line 135. This sentence is repetitive to a similar line in your introduction on line 95 so I would delete it here. I also recommend deleting the sub-section of pre-pandemic and pandemic phrase as these are very short and these definitions can both sit together under the wider sub-section heading of ‘2.2 Definition of the pandemic phases’
Line 151. I am not clear what 11 out of 20 monthly visits means? That 20 visits (one of a month) were made during this period but pellets were only collected during 11. Or that only 11 visits were made during this period? Please clarify. And similarly on line 157.
Line 169. Please include the definition of macroplastic pieces i.e. all pieces over x mm in size? You also don’t mention in your methods how you classified colour? What reference categories/standards did you use (i.e. see Provencher et al. 2017 on standardised ways of classifying plastics) as classifying colour can be subjective based on the observer.
Line 172. To be consistent throughout the manuscript I recommend using debris rather than rubbish, and use the tern non-plastic debris when not referring to plastics. The line on 190 can then be changed to ‘Debris was found in 19……’. And material changed to debris on line 202.
Line 179. So this is the proportion of plastic items of all anthropogenic debris items identified? Please clarify.
Line 190. Please include over the entire study period (i.e. during both the pre-pandemic and pandemic phrases) if I have understood this correctly?
Line 191. Here you say 147 items were classified as plastics but the total for the ‘Plastic items’ column in the Supp. Material data is 195. Can you check / explain this discrepancy.
Line 202. As above, the total items in your ‘Count MA (rubbish)’ column is 5? And there is no mention of balloon in the this Supp. Material data.
Section 3.2. This section is very descriptive and basically repeats what is in Figure 4. It would be useful to focus this on the temporal variation that Figure 4 shows in both phases and just pick out the key points i.e the peaks and lows. These results would be stronger by also accounting for month in your analysis – given that there does appear to be a temporal pattern across the year even ignoring the pandemic phase, I assume related to peak tourist seasons? Put this would be useful to clarify.
Line 204. It is not clear what you mean by threadlike here, please clarify. Threadlike debris that was not plastic? If so could you identify what material this was?
Line 216. In 5 additional pellets to those containing plastics? Or did these pellets contain both plastics and non-plastic debris?
Lines 221 – 234. This is the first time differences in the lockdown restrictions and therefore different intensity of pellet collection has been mentioned – this should be introduced during your methods i.e. in the ‘2.2 Definition of the pandemic phases’ section. And state here that you look at the FO% of debris in plastics over time during the different phases. I see that these different periods are included in Table 1 so that should be more clearly stated in the main text i.e. state in the main text that Table 1 describes the different lockdown periods.
Figure 4. This is a really informative figure. It also highlights the temporal variability in the % of plastics in pellets across the year both before and during the pandemic. Although on a whole less plastics were consumed during the pandemic period the peak in February 2021 (during a strict partial lockdown) is similar to that in January 2020. I guess this time of year coincides with the peak tourist season compared to July for example? Although it is also interesting that the % was so low in Feb/March 2020? Please acknowledge this temporal variability in your discussion and what might be driving this. There will also be a time lag before the Cormorants ingest the plastics as first need to enter the ocean, break down and be consumed by the fish – which may in part explain the unexpected peak in January 2020. But this is another limitation that needs to be addressed.

·

Basic reporting

some issues with data reported (%N not useful) - see general comments under section 4

Experimental design

It would be greatly improved if there were post-COVID data showing a concomitant increase in litter in pellets. Given >2 years since restrictions ended, this does not seem to be an unreasonable expectation

Validity of the findings

see general comments under section 4

Additional comments

This paper reports a reduction in the proportion of cormorant pellets containing litter items during the COVID-19 lockdown in Lima. It is certainly an interesting finding, but in addition the paper needs to 1) test for an impact on total litter loads, and 2) outline the mechanism proposed to drive this result. These points are expanded on below:

1) The paper needs to report total litter loads in pellets (the number of plastic and other anthropogenic items combined), and to test whether these loads also vary with lockdown levels. The metric %N is not very helpful – as I understand it, it just reports the proportion of ingested debris which is plastic. Currently, the only data supporting the impact of COVID on plastic in cormorant pellets is the change in %FO presented in Fig. 4. Overall, a chi-square test shows a drop in the %FO, but the data shown in Fig. 4 are quite noisy, and additional supporting evidence would be very helpful.

2) The paper fails to clearly establish a link between the reduction in plastic in cormorant pellets and the COVID restrictions on human movements. The Discussion implies that by restricting beach access there was less litter entering the sea, but the kinds of items ingested by the cormorants are not typical of beach-goer litter. From the images of litter items, they are mostly fragments of items which have been in the environment for some time, indicating you would expect a lag effect between the reduction in litter inputs and a reduction in litter in cormorant diet. My study in South Africa of how street litter loads responded to easing of lockdown restrictions showed an immediate response to easing of restrictions (Ryan et al. 2020, Environmental Processes 7: 1303-1312, doi: 10.1007/s40710-020-00472-1), but the beach litter sampling that we conducted during the lockdowns showed no change in the amounts of litter washing ashore, presumably because of lags in the system.

Exacerbating the lag in this study is the fact that most if not all items are ingested secondarily from contaminated prey. In order for the change observed in cormorant pellets to be caused by the COVID lockdown there would have to be very rapid turnover in the standing stocks of litter in the coastal waters off Lima. Maybe this is the case, given strong upwelling in the region, but this process needs to be a central point of the Discussion.

The paper concludes “Future research should focus on the assessment of the occurrence of plastic in cormorant pellets after the pandemic”. Given that the lockdowns ceased several years ago, I would be much happier if there were a third set of data from the post-COVID period showing that litter loads increased again to pre-COVID levels – this would greatly strengthen the case that the lockdown was directly responsible for the observed reduction.

The conclusions also notes that future research should investigate “the polymer type ingested using Fourier-transform Infrared spectroscopy and examine any gastrointestinal damage in fresh carcasses”. These points are not directly relevant to the current study. There is no mention of dead cormorants.

Several other points are noted directly onto the pdf of the manuscript.

---

## Round 0.2 · Major Revisions

· Academic Editor

Major Revisions

Thank you for efforts in revising your article in response to the comments received on the previous version. I sent your revised manuscript back to the original reviewers and now received comments from one of them (the other was unavailable). They still perceive some of the issues that were previously raised as too fundamental to accept the manuscript in its current form.

I agree that some of the conclusions (for example, the final sentence of the abstract) are still stated without providing sufficient underlying information to fully assess whether the correlation you detect can lead to them. As the other reviewer pointed out in the previous assessment, it would help if you more clearly describe the hypothesis in the introduction. Having a clearer explanation of the potential pathway you are testing, plus specific predictions, could help to guide further research and clarify the limitations. You also do not sufficiently acknowledge that alternative factors could explain the result. For example, from the top of my head, maybe the cormorants changed their foraging behavior during the pandemic, feeding on prey that was previously less accessible and that is less contaminated. As the reviewer, I do think that the data and results are sufficiently substantial and interesting for a publication. The issue is about the interpretation of the results, and that you focus on one particular interpretation without having sufficient evidence that the underlying assumptions are met. The reviewer provides more detailed suggestions for how to change some of the statements in the different parts of the article. I hope these will be helpful for you when you revise the article.

Reviewer 1 ·

Basic reporting

The manuscript is generally clear with the raw data shared and adequate structure and figures/tables.

Experimental design

The hypothesis is a now little vague in what the authors were expecting - given the change in how the authors have interpreted their findings. The introduction (and abstract) focuses on the high number of tourists that visit the area and how this is linked to high plastic pollution, which differs from the discussion where the focus in on how these tourists might disturb plastics already present in the environment/beaches.

Validity of the findings

Although the authors found that significantly more pellets contained debris during the pre-pandemic phase compared to the pandemic / lockdown phase, I am not convinced by the discussion interpreting this difference. Are there other studies that show that human disturbance increases the amount of micro-plastics transferring from beaches into the sea for example?
For example:

Line 277-293. I am really not sure I am convinced by this argument - although I agree it is not impossible. My main concern is that there will still be a lag between these microplastics disturbed from the beach getting into the sea and being ingested by prey fish/species that the cormorants then feed on. As you state on line 94 and 356 - Cormorants will predominantly be ingesting plastics through secondary ingestion via their prey species. The time lag between plastics entering the sea and being consumed by prey species is therefore still not adequately discussed/accounted for.

Line 293 - 303. It is not clear how this strengths / backs your argument, as this coastal erosions / plastic discharge from rivers would occur even in a lockdown especially the coastal erosion if not caused by direct human activity?

Line 358. I feel it is unlikely that cormorants would be accidently ingesting plastic items in the water column, at least to any great extent given that they are pursuit hunters. Therefore, I am not convinced by this argument. Furthermore, if cormorants were accidently ingesting plastics directly, I would assume these would be fish sized items, rather than microplastic sized as the authors found in the pellets? Typically, in pellet producing species like cormorants, larger inedible items such as plastics will be expelled in pellets rather than digested.

The points made in the discussion around human activity on beaches being responsible for accelerating the presence of plastics in the sea and therefore available to cormorants and their prey are not reflected in the abstract or in the introduction where the focus instead is on less plastic entering the environment as a whole due to changes in human behavior associated with the lockdowns.

Line 412. Although there is a link/correlation I am not convinced of the cause and effect of less plastics in the cormorant pellets being attributed to changes in human activity during the different phases. Although, it is potentially convincing to say it was the lockdown and reduced human activity resulted in less plastics in the cormorant pellets - the authors can't rule out other external influences that were not measured, especially as the backgrounds levels of annual variation in FO% of plastics in the pellets was not known pre-pandemic.

Additional comments

Thank you to the authors for addressing most of my previous comments - and I commend the authors for thinking outside the box to try and explain the differences observed in the FO% of plastics found in the cormorant pellets between the two phases. I just feel that more evidence is required to make this arguement a convincing one.

The abstract has not changed to reflect the new argument of people's activity on the beaches / entering the sea resulting in more plastic in the oceans outwith the lockdowns.

Line 140. Not clear what you mean here by close immediately after the lockdown - i.e. access was permanently removed?

Line 180. It is not clear what you mean here by 'plastic debris (>0 mm) were measured'? It is impossible for an item to be 0 mm. Do you mean something like > 0.1 mm? Please clarify.

Line 194. It is unclear to me what the Litter load is measuring? i.e. if you count 120 debris items in 20 pellets during a single month = 6? Please clarify.

Line 234. This sentence appears unfinished?

Figure 2. Include in the figure legend than the number at each point refer to the number of pellets this % occurrence is based on.

Figure 3. It is not clear from this figure what Litter Load refers to? This should be explained in the figure legend.

Line 348. This paragraph does not flow well from the previous paragraph. It is not obvious how this first sentence is relevant at this point based the rest of the paragraph.

---

## Round 0.3 · Minor Revisions

· Academic Editor

Minor Revisions

Thank you for making these additional changes to your manuscript. The additional analyses you added provide valuable further insights, and I appreciate that you brought in another expert as a co-author for these analyses. The revisions to the abstract and discussion now also make it clearer what your study contributes and what additional information would be needed to further understand the patterns you describe.

I only have a few minor comments in relation to the additional text you added during this last revision (line numbers refer to the tracked changes manuscript). These are only minor suggestions for changes to the phrasing, after which your manuscript should be ready for acceptance.

Abstract, Line 37: The sentence "Additionally, our study highlights the direct correlation between human presence on beaches and the quantity of microplastics (mainly threadlike) found in cormorant pellets" repeats information that is already presented in the previous sentence. In addition, this sentence still makes it appear as if the there is a direct link between human presence on the beach and plastic in the pellets, whereas your discussion now integrates the feedback from the reviewers that many factors changed during the pandemic, all of which could have contributed to the observed changes. I suggest you remove this sentence.

Methods, Line 159: the explanation of the sampling along the median strip now appears to contain some repetition, where the newly added text restates what the not-deleted previous text also appears to explain.

Methods, Line 219: While you provide more explanation in the response to the reviewer as to what "total litter load" represents, I think it would be helpful to also clarify this in the text. I also wonder whether the term should be changed. Based on your description, it is not the total litter load (as in the sum), but the average litter load (items divided by total pellets). Similarly, in Figure 3, I think it would help to specify that the litter load is the "average number of items per pellet". Right now it is slightly confusing because the y-axis only has values smaller than 1, while the term number would imply a count of 1 or larger.

Results, Line 293: you provide a comparison of the sizes of plastics found before and during the pandemic. However, in the abstract and discussion you focus on the type of plastic found ("threadlike"). Can you perform a similar assessment to see whether the proportion of threadlike plastic relative to the two other categories (sheetlike,fragments) changed from before to during the pandemic?

Discussion, Line 305: While your additional analyses now refer to the diet of your study species, the discussion still does not mention that the most likely source of plastic in the pellets is "microplastics disturbed from the beach getting into the sea and being ingested by prey fish/species that the cormorants then feed on" (quote from the text of the reviewer from the previous round). I think it would help to mention how the cormorants most likely ingested the plastic, because this appears crucial for any speculation about the potential reasons for how the changes during the pandemic might have influenced the result you observe. This is in particular helpful for readers like me, who are not as familiar with the behaviour of cormorants.

---

## Round 0.4 · accepted · Accept

· Academic Editor

Accept

Thank you for addressing my remaining comments so diligently and for making the respective changes in the manuscript. This addresses all points previously raised by reviewers, and I am happy to recommend acceptance of the article.